# Advanced Biological Oxidation of Domestic Sewage with the Use of Compost Beds in a Natural Treatment System for Wastewater

Wojciech Halicki [1,2]

1   Institute of Applied Ecology, Skórzyn 44a, 66-614 Maszewo, Poland; w.halicki.ies@gmail.com
2   Centre for East European Studies, Faculty of Oriental Studies, University of Warsaw, ul. Krakowskie Przedmieście 26/28, 00-927 Warsaw, Poland

**Abstract:** Due to the progressing problems with ensuring sufficient quantity and quality of water for municipal, irrigation and economic purposes, the pressure to reuse treated wastewater is increasing. This fact forces the development of advanced systems enabling more effective wastewater treatment. This article presents the results of a 2.5-year study period in which compost beds, which are part of a natural treatment system for wastewater (NTSW), were used to treat domestic sewage by fully removing easily degradable organic matter and by fully nitrifying ammonium nitrogen. It was shown that the compost environment provides complete access to oxygen for the coexisting heterotrophic and autotrophic bacteria, covering 100% of their oxygen demand. Moreover, the outflow of treated wastewater shows an oxygen content of 4–7 g $O_2/m^3$. Advanced biological oxidation occurring in the compost beds with an area of 1 $m^2$ per inhabitant and a daily hydraulic load of about 100 $L/m^2$ can effectively and without additional energy expenditure provide a 98% reduction in biological oxygen demand and a 99.5% reduction in ammonium nitrogen. In addition, the effluent from the compost filters meets the most stringent quality criteria for (1) treated wastewater used for irrigation and (2) bathing water in terms of microbiological contamination.

**Keywords:** advanced oxidation; advanced technologies; natural treatment system for wastewater; water renewal; nitrification; oxygen supply; compost beds; microbiological contamination

## 1. Introduction

According to the Food and Agriculture Organization of the United Nations, the greatest challenge of the 21st century is water scarcity [1], and by 2025, approximately 1.8 billion people will be living with absolute water scarcity [2]. Today, about half of the world's population already lives in areas of potential water scarcity, and this number may increase to about 5.7 billion by 2050 [3]. The future challenge for water management is not only related to the quantity of water resources available on our planet but also to water quality [4]. Surface water is the main source of water supply for human needs, but its quantity and quality are continuously deteriorating. [5]. This is due to the dynamic growth of the world's population, combined with rising standards of living, increasing water consumption in agriculture and industry and increasing water pollution from untreated wastewater discharges [6]. For this reason, we will observe the increasing importance of groundwater as the most secure and drought-resistant source of drinking water supply worldwide. This scenario will occur despite the fact that one-third of the world's largest groundwater systems are already in distress due to over-exploitation and pollution [7]. Although this problem of water scarcity and quality first affects the regions of Asia and Africa, it is now also occurring in Europe. In the European Union, as much as 24% of groundwater and 22% of surface water is now of poor quality [8,9].

Considering the increasing water demand, water for human activities and the continuous decline in the quantity and quality of surface and groundwater, proper water management must play a crucial role now and in the future in order to keep water demand

at a sufficient level. This goal must be started locally and then globally. To meet local and global needs, water management must be able to implement new advanced technologies and new approaches to wastewater treatment and reuse, reduce fresh water demand and limit the amount of pollution discharged into surface and groundwater. Most of the water needed for agriculture, industry and domestic use does not have to be of potable quality. Some of this water can be supplied by recycled water from wastewater treatment plants, leaving fresh water sources for potable use only. In addition, the approach to wastewater management needs to change. To achieve sustainability in water management, wastewater must be treated as a resource, not as a waste.

The other reason for the increasing challenge of sustainable water management is the need to use advanced technologies because more and more contaminants can be identified in wastewater, fresh water and drinking water [10]. Using advanced technologies to treat wastewater allows us to successfully remove a wide range of different pollutants from wastewater and to protect water resources from undesirable components [11]. According to the American Institute of Chemical Engineers, advanced wastewater treatment is defined as any process that reduces the level of pollutants in wastewater below the level achievable by conventional secondary or biological treatment [12]. This process includes the removal of nutrients such as phosphorus and nitrogen, as well as suspended solids and microbiological contamination. Advanced water and wastewater treatment involves the use of biological or physicochemical processes. In general, the advanced biological process is responsible for the removal of nutrients and, to a limited extent, a higher reduction in natural organic matter. Physicochemical processes are used to remove total suspended solids, natural and synthetic organics, nutrients and disinfection (removal of microbiological contamination). For greater efficiencies in wastewater treatment, both processes are used together, as in the case of wastewater treatment to achieve drinking water quality. The application of new advanced technologies depends on a number of limiting factors, such as investment and operating costs, demand for recycled water and nutrients from wastewater and public awareness. Therefore, the application of advanced technologies will be limited to large treatment plants and relatively rich societies. More benefits could be achieved using the natural treatment system for wastewater (NTSW), which can be successfully implemented in the same way as advanced technical wastewater treatment plants, including membrane filtration systems, automatic variable filtration (AVF), advanced oxidation processes (AOP) or ultraviolet (UV) irradiation.

Recent publications indicate that some types of NTSWs can purify domestic wastewater to drinking water quality [13,14]. In this case, the efficiency of this system has reached the highest level of wastewater treatment. It can be stated that most NTSWs are able to achieve higher pollution reduction than conventional secondary treatment systems [15,16]. In addition, properly designed, constructed and operated NTSWs can simultaneously treat wastewater, renew water and are themselves an ecological use of water by creating new habitats [17]. The next important factor is that NTSWs have been shown to be suitable for industrial, agricultural and domestic wastewater treatment [18–20]. In addition, the most recent experience indicates the possibility of using natural methods in the phytoremediation processes by treating textile wastewater [21]. Considering all the benefits, this type of treatment system should no longer be treated as a conventional secondary treatment system but rather as an advanced technology. This system could play a very important role in the decentralized water management system in the future by enabling full water reuse. In most cases, the use of an NTSW will result in a system that is less expensive to construct and operate and requires less energy than conventional mechanical treatment alternatives [22]. One of the most limiting factors in the application of NTSWs is the relatively high demand for land area [23]. The reason for this is the insufficient supply of oxygen to the beds or to the bodies of water (in the pond treatment systems), which leads to the prolongation of the decomposition processes of organic pollutants and the oxidation of nitrogen. For this reason, achieving satisfactory results in cleaning the NTSW requires a relatively high surface area.

This study describes the performance of an NTSW pilot plant over 2.5 years of continuous operation with the aim of (1) achieving effluent water with potable water quality, (2) enabling a complete, versatile water reuse that is safe for both people and the environment, (3) reusing substances contained in the effluent to produce compost and (4) reducing the surface area of the NTSW while maintaining the highest efficiency of purification. The pilot plant consists of compost beds for wastewater treatment and water renewal beds. The performance of the pilot plant in terms of the removal efficiency of organic matter and biogenic substances has already been evaluated and described in [13,14]. The main concern presented in this article focuses on the efficiency of the following:

- The oxygen supply in the compost beds, which is responsible for the complete oxidation of organic matter and ammonia nitrogen,
- The successful removal of microbiological contamination,
- The production of good-quality compost.

## 2. Materials and Methods

### 2.1. Description of the NTSW Pilot Plant

This research was carried out on a pilot plant consisting of two parts, one for wastewater treatment and the other for water renewal. This NTSW plant was part of a research project aimed at treating domestic wastewater to a level similar to drinking water. A detailed description of the whole plant can be found in Halicki and Halicki [13]. So far, two papers have been published on this NTSW plant [13,14]. The first publication described the effect of the removal of organic compounds and the elimination of total suspended solids, while the second publication described the effect of the removal of nitrogen and phosphorus. This publication focuses on the oxidation process that takes place in compost beds, which provide good conditions for advanced biological oxidation. The final publication in this research project is currently being written. It focuses on the elimination of pathogenic microorganisms in this pilot plant. During this 2.5-year research study, wastewater and reclaimed water were analyzed for the following parameters: biological oxygen demand (BOD), chemical oxygen demand (COD), oxidation, total suspended solids (TSS), total nitrogen (TN), $NH_4$, $NO_3$, $NO_2$, total phosphorus (TP), $PO_4$, pH, conductivity, oxygen, K, Cl, Mn, Ca, Fe, coli bacteria, enterococci and colony count 22 °C.

The following description focuses on the first part of this pilot plant, which consists of a septic tank and compost beds. The pilot plant was built on the premises of the Institute of Applied Ecology (IES, http://ies.zgora.pl/en/home/, accessed on 17 July 2023) and is designed to treat sewage from the IES building with 3 residents and up to 5 office workers. The institute is located in western Poland (52°7′15.808″ N, 15°2′21.102″ E). The compost part of this pilot plant is divided into three separate beds operating under different daily hydraulic loads, namely 0.07, 0.1 and 0.13 $m^3/m^2$ for beds A, B and C, respectively (Figure 1). Photographs of the pilot plant are shown in Figure 2.

The septic tank has a capacity of 3 $m^3$ and receives an average daily wastewater load of 300 L from the IES building. The retention time of wastewater in this tank is about 8 days, during which physical (sedimentation of suspended solids) and partially biological (anaerobic decomposition) wastewater treatment takes place. Every two years of operation, the septic tank is emptied of the stored sludge. The compost beds, which are responsible for the treatment process, are located in a greenhouse and are divided into 3 separate parts (A, B and C) with an area of 1 $m^2$ each. Each bed is 1 m high and consists of the following layers: from the bottom, a 10 cm layer of gravel (4 to 16 mm granulation), a 10 cm layer of sand (0.2 to 2 mm granulation) and an 80 cm layer of compost (80% wood chips and 20% peat soil mixed with 10 kg of fertilizer lime containing 60% calcium carbonate).

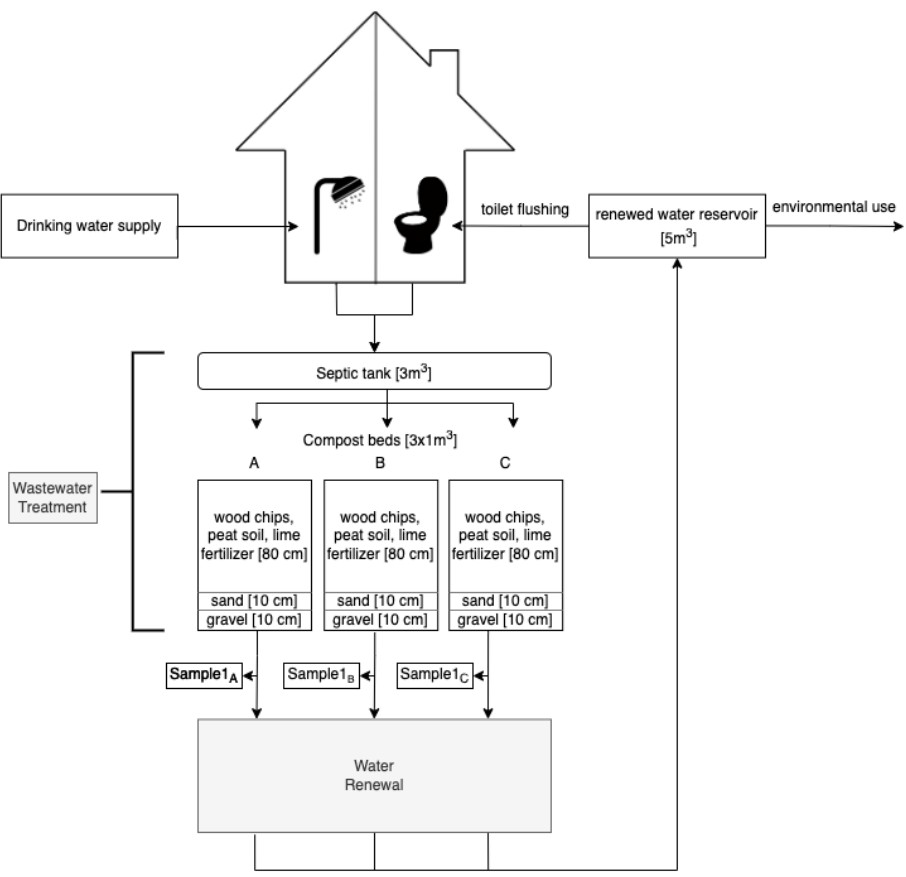

**Figure 1.** Scheme of the NTSW pilot plant (adapted from Halicki and Halicki [13]). The three parallel wastewater treatment processes are operated at different daily hydraulic loads, namely 0.07, 0.1 and 0.13 m$^3$/m$^2$ for the letters A, B and C, respectively.

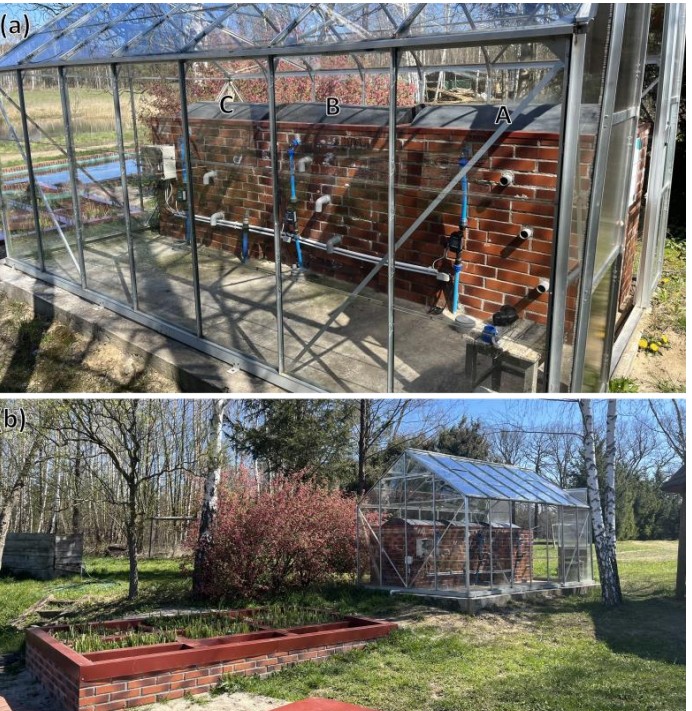

**Figure 2.** The NTSW pilot plant (adapted from Halicki and Halicki [13]). (**a**) Compost beds (A, B, C) in the greenhouse. (**b**) Installation of the whole NTSW pilot plant with part of the water renewal.

### 2.2. Sampling and Quality Assessment

Sampling of the treated sewage in the effluent from the compost beds took place on average every two weeks. Raw sewage sampling was performed on average once a month. In total, 48 and 92 samples were collected of the raw sewage and the effluent from the compost beds, respectively. The time span of this study ranges from August 2019 to May 2022. The analyses of BOD and ammonia nitrogen compounds were performed according to the Standard Methods for the Examination of Water and Wastewater [24]: for ammonia nitrogen, the method designated as 4500-NH3 E, for BOD of raw sewage, the method designated as respirometric method (5210 D), and for BOD of treated sewage, the method designated as 5-day BOD test (5210 B). The bacteriological analyses of the effluent from compost filter A were carried out between May 2021 and June 2022 in 16 and 15 measurements for *Escherichia coli* and enterococci, respectively. Both analyses were performed according to the methods of ISO 7899-1:1998 [25] and ISO 9308-3:1998 [26]. The nutrient analysis of the compost (compost filter A) was performed in May 2023 (4 years after the start of operation of the NTSW) by a certified laboratory (Regional Chemical and Agricultural Station in Gdańsk, accreditation standard: PN-EN ISO/IEC 17025:2018-02 [27]).

### 2.3. Wastewater Treatment Process

The compost beds in this pilot plant perform two important functions related to advanced technology. First, the removal of organic matter and nitrification, which consumes oxygen, takes place in these three compost beds. The second task is the production of compost. Compost beds allow wastewater to be used as a nutrient source to produce compost as a natural, valuable organic fertilizer. These compost beds work similarly to traditional composters but with the following differences:

- In traditional composters, living organisms decompose various solid wastes and produce compost (natural fertilizer). In the compost beds of this pilot plant, the living microorganisms decompose mostly dissolved organic matter that flows into the compost beds from the septic tank.
- The compost beds are continuously fed with organic matter every hour, while in traditional composters, this happens occasionally.
- The compost beds are filled with wood chips, while the traditional composters are filled with various solid wastes.
- In traditional composters, oxygen is used primarily for decomposition of organic matter, while in compost beds, it is used primarily for nitrification and, to a lesser extent, for decomposition of organic matter contained in wastewater.
- In compost beds, the filling material becomes valuable compost only after at least three years of use, while in traditional composters, it becomes valuable after one season (a few months).

The average hydraulic retention time (HRT) in the compost beds is approximately 1 h. The HRT indicates the duration (in hours or days) that wastewater remains in the treatment plant before it is discharged. In this short time, the organic matter and ammonium nitrogen contained in the wastewater are first adsorbed on the surface of the compost and then decomposed. In practice, this decomposition time can be longer than 1 h. Aeration of the compost beds occurs naturally by diffusion through pores in the compost material. Inside the compost beds, a rich environment of typical soil/compost fauna and flora is created [28]. Heterotrophic and autotrophic bacteria, actinomycetes, fungi, protozoa, nematodes, vases, earthworms and others are involved in the purification process in the compost beds [29]. In addition, compost has favorable sorption properties for hardly degradable organic substances, which contributes to the retention of these substances in the beds and facilitates their slow mineralization. After passing through the compost layer, the treated wastewater seeps through the sand layer, where the fine suspension washed out of the compost is removed. The gravel layer on the bottom allows the treated wastewater to drain from the beds.

The purification process in the compost beds takes place in several stages. In the first stage, heterotrophic bacteria decompose the organic matter dissolved in the wastewater. This process releases carbon dioxide and various mineral substances such as nitrogen, ammonia, phosphate, sulfate and others. In addition, the decomposition of organic matter results in the growth of heterotrophic bacteria that produce excess sludge in compost beds. In the second stage, autotrophic bacteria oxidize most of the ammonia nitrogen to nitrate nitrogen. This process requires much more oxygen than the process of decomposition of organic matter. The released nitrate nitrogen is subject to another process called denitrification, which also occurs in facultative aerobic conditions in compost beds. In the next stage, other organisms living in the compost consume and mineralize the excess sludge. The fungi living in the compost beds are constantly active in decomposing the wood chips with the help of nutrients from the wastewater, which results in the release of humus substances. All these processes taking place in compost beds are very dynamic and effective and contribute to excellent purification of wastewater and production of valuable compost. The concept of this pilot plant assumes that the compost contained in the beds can be replaced every 3 or 4 years with fresh wood chips mixed with peat soil. The compost obtained in this way can be used as a natural fertilizer. One of the most important factors enabling such successful cooperation of different organisms working in the compost condition is the sufficient oxygen supply.

## 3. Results

### 3.1. Oxygen Demand

Table 1 shows the oxygen demand (average, maximum, minimum, standard deviation (SD) and coefficient of variation (CV)) in the raw wastewater flowing from the septic tank to the compost beds during the entire project period. The value of oxygen demand for ammonium nitrogen nitrification is based on the stoichiometric equations:

$$NH_4^+ + 1.5O_2 \rightarrow 2H^+ + H_2O + NO_2^-$$
$$3.21 \text{ mg } O_2/\text{mg NH}_4^+\text{-N}, \tag{1}$$

$$NO_2^- + 0.5O_2 \rightarrow NO_3^-,$$
$$1.11 \text{ mg}O_2/\text{mg NO}_2\text{-N}, \tag{2}$$

as presented by Rheinheimer et al., 1988 [30].

**Table 1.** Oxygen demand in raw wastewater flowing from the septic tank to compost beds.

| Oxygen Demand (g $O_2/m^3$) | Mean | Max. | Min. | SD | CV |
|---|---|---|---|---|---|
| BOD | 232.1 | 410.0 | 53.1 | 91.9 | 40.0 |
| NH$_4$-N | 621.1 | 1453.3 | 301.6 | 241.6 | 38.8 |
| Sum | 853.3 | 1863.3 | 354.7 | 275.3 | 32.3 |

The results presented in Table 1 indicate that the main oxygen demand in this wastewater is necessary for the nitrification of ammonium nitrogen. Generally, in this case, the oxygen demand for the oxidation of ammonium nitrogen constitutes from 72% to 85% of the total oxygen demand in this raw wastewater. The high SD and CV of the oxygen demand in the raw sewage are due to the fluctuation in the number of occupants and employees present in the IES building.

The efficiency of oxygen demand reduction in the compost beds, presented in Table 2, is at the same time an indicator of the oxygen supply to these beds, which is necessary for the purification process. This means that the higher the reduction in oxygen demand in compost beds, the better the oxygen supply to these beds, and finally, the purification processes run more efficiently. In other words, the reduction in oxygen demand in compost beds is directly proportional to the removal of organic matter and the nitrification of ammonium nitrogen. In spite of different hydraulic loads, the reduction in oxygen demand

in all beds is over 97%. The achieved reduction values (from 97 to 99.6%) are very stable in this range and generally depend in a small span on the daily oxygen demand fluctuating from 24.8 to 242.2 g $O_2/m^2$. The most stable reduction was achieved in compost bed A with a daily hydraulic load of 0.07 $m^3/m^2$. Regardless of different daily oxygen demands ranging from 24.8 to 130.4 g $O_2/m^2$, the reduction effect in compost bed A was constant and above 99%. Despite higher hydraulic load in the other beds (B and C), even greater reduction (up to 99.5 and 99.6%) was achieved in these beds for the minimum daily load of oxygen demand, accounting for 35.4 and 46.1 g $O_2/m^2$ for B and C beds, respectively. The data presented in Table 2 refer to the average results of the whole study period. The high values of CV, especially in the outflow, are caused by very low absolute outflow concentration values. However, the SD values are very low, namely 0.45, 2.35 and 1.89 g $O_2/m^2$ for the A, B and C beds, respectively. Therefore, it is clear, that the general variation in oxygen demand in the outflow is low.

**Table 2.** Daily oxygen demand for the oxidation of easily decomposable organic matter (BOD) and ammonium nitrogen in the influent to and effluent from the compost beds.

| Compost Bed | Value | Inflow (g $O_2/m^2$) | Outflow (g $O_2/m^2$) | Reduction (%) |
|---|---|---|---|---|
| A<br>0.07 $m^3/m^2/d$ | mean | 58.4 | 0.42 | 99.3 |
|  | max. | 130.4 | 1 | 99.2 |
|  | min. | 24.8 | 0.08 | 99.7 |
|  | SD | 19.19 | 0.45 | - |
|  | CV | 32.9 | 107.1 | - |
| B<br>0.1 $m^3/m^2/d$ | mean | 85.3 | 0.97 | 98.8 |
|  | max. | 186.3 | 4.4 | 97.7 |
|  | min. | 35.4 | 0.13 | 99.6 |
|  | SD | 26.83 | 2.35 | - |
|  | CV | 31.5 | 242.3 | - |
| C<br>0.13 $m^3/m^2/d$ | mean | 110.9 | 1.4 | 98.7 |
|  | max. | 242.2 | 3.6 | 98.5 |
|  | min. | 46.1 | 0.2 | 99.5 |
|  | SD | 35.38 | 1.89 | - |
|  | CV | 31.9 | 135.0 | - |

Figure 3 shows the average quarterly results for the entire study period. In Figure 3a, we can see short data gaps caused by sampling errors. In this first graph (Figure 3a), there are two characteristic periods, the first from August 2019 until February 2020 and the second from April 2020 until April 2022. In this first period, the oxygen demand was about 100% lower than in the second period. The second, much longer period is characterized by frequent changes in oxygen demand in the influent to the compost beds. Despite these changes, the effluent of all the beds is characterized by a much more stable value (Figure 3b). The outflow of the first bed (A) shows the smallest change, which means that the compost bed with a daily hydraulic load of 0.07 $m^3/m^2$ is able to withstand periodic increases in oxygen demand load in the inflow and maintain very stable values of oxygen demand in the outflow.

Figure 4 shows the seasonal variability of the oxygen demand in the influent (Figure 4a) and the effluent (Figure 4b) of the compost beds. The oxygen demand in the influent of the compost beds is significantly higher in the first six months than in the second half of the year (Figure 4a). The load of oxygen demand in compost bed A is effectively reduced throughout the year. On the contrary, in compost bed B and especially in bed C, a decrease in the reduction efficiency can be observed during the summer months. One of the reasons for the lower efficiency in summer months could be a decreasing oxygen supply to these beds. As can be seen in Figure 5, which shows the oxygen concentration in the effluent of all compost beds, there is a continuous presence of oxygen throughout the year. These results confirm that the oxygen supply to the beds was sufficient throughout the study

period. Therefore, the decrease in oxygen demand in the summer months in beds B and C was not caused by the lack of oxygen but by other factors.

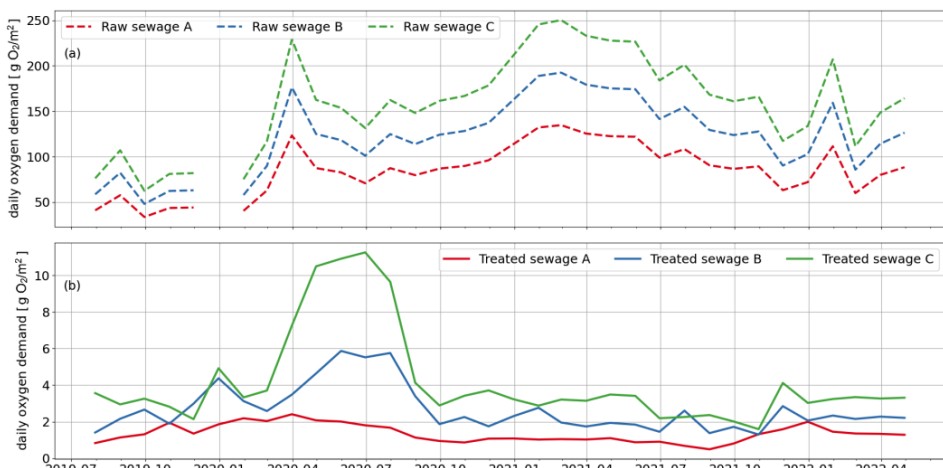

**Figure 3.** Average daily oxygen demand: (**a**) inflow to the compost beds, (**b**) outflow from the compost beds.

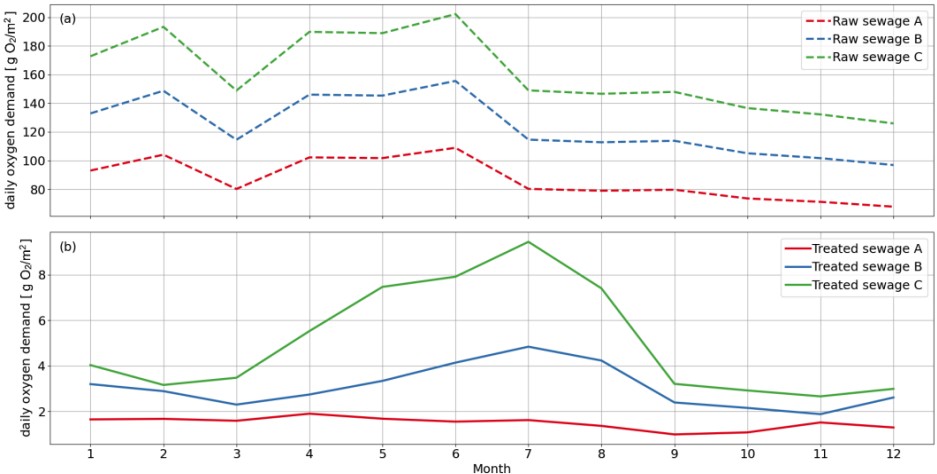

**Figure 4.** Seasonal variation in oxygen demand load: (**a**) in the influent of compost beds, (**b**) in the effluent of compost beds.

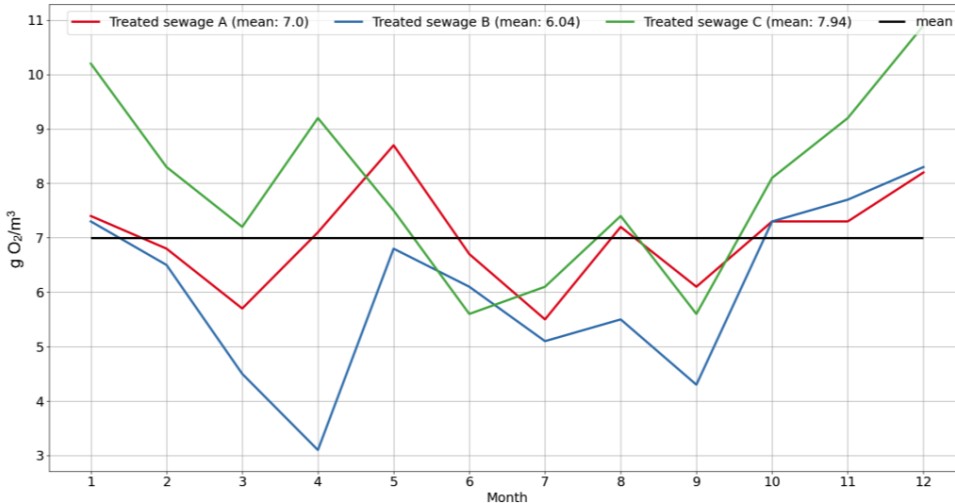

**Figure 5.** Seasonal oxygen concentration in treated wastewater from three compost beds.

### 3.2. Microbiological Contamination

The average number of *E.coli* bacteria in the outflow of the compost filter A is 256 cfu and ranged from 18 to 818 cfu during the study period. For the number of enterococci, the mean, minimum and maximum values were 61, 4 and 245 cfu. Figure 6 shows the distribution of analysis results for enterococci (a) and *E. coli* (b). The three categories in each figure (the (1), (2) and (3) numbers below the bars) refer to the European requirements for the maximum number of bacteria in 100 mL of bathing water recommended for excellent transitional water quality (1), excellent inland water quality (2) and good inland water quality (3) [31]. It is apparent from Figure 6 that the number of bacteria in the outflow of the compost filter A meets the criteria for excellent bathing water quality for transitional waters in 86% (13 of 15 samples) and 62% (10 of 16 samples) of samples for the enterococci and *E. coli*, respectively. Furthermore, only 1 and 3 measurements (out of 15 and 16, respectively, for enterococci and *E. coli*) showed a number of bacteria within the criteria of good bathing water quality for inland waters. The rest of the measurements showed a number of bacteria within the excellent transitional or inland bathing water quality criteria.

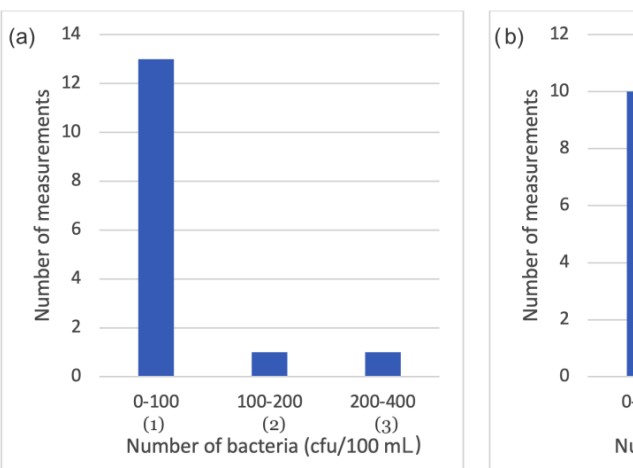 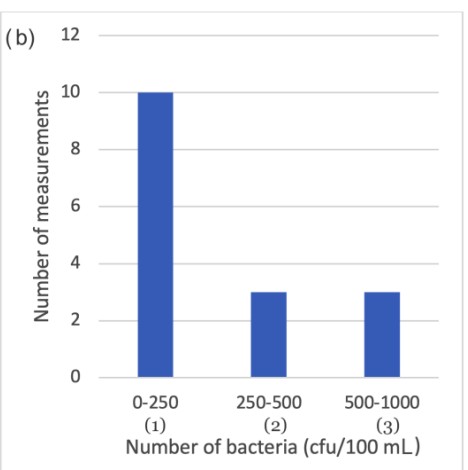

**Figure 6.** Bacteriological analysis results from the compost filter A for (**a**) enterococci and (**b**) *E. coli*.

### 3.3. Nutrient Content of Compost

Two samples were taken from compost bed A and analyzed for the main nutrients such as nitrogen, phosphorus, potassium, calcium and magnesium. The results of this analysis are shown in Table 3. During the four years of operation of the composting beds, the initial material (80% wood chips and 20% peat soil mixed with 10 kg of fertilizer lime containing 60% calcium carbonate) was transformed into valuable compost. As can be seen from Table 3, the compost contains nitrogen, phosphorus and magnesium in concentrations typical of natural fertilizers [32]. In contrast, the amount of potassium is very low, which is related to its generally low concentration in the wastewater. Finally, the tested compost samples have a very high calcium content, which is due to the addition of calcium carbonate to the wood chips.

**Table 3.** Nutrient content of compost samples from compost bed A. Sample one was taken from the top layer of the compost (1 to 10 cm deep), and sample two was taken from a layer 10 to 20 cm deep.

| Parameter | Sample 1 | Sample 2 | Unit | Uncertainty |
|---|---|---|---|---|
| Dry weight in fresh weight | 31.6 | 34.2 | % | ±5% |
| Total nitrogen | 1.11 | 1.20 | % of dry weight | ±10% |
| Total phosphorous | 0.16 | 0.20 | % of dry weight | ±15% |
| Potassium | <0.10 * | <0.10 * | % of dry weight | ±15% |
| Calcium | 6.71 | 6.73 | % of dry weight | ±22% |
| Magnesium | 0.17 | 0.20 | % of dry weight | ±15% |

* Result below the lower limit of the measuring range of the method.

## 4. Discussion

One of the hypotheses of this paper was that the pilot plant described in this study could be considered an example of advanced biological oxidation and advanced technology. However, as of today, there is no such concept as advanced biological oxidation. In practice, the concept of advanced oxidation is used, which refers to a chemical process defined as the oxidation processes involving hydroxyl radicals (OH·) and sulfate radicals ($SO_4^- \cdot$) to improve the water purification process [33]. In this sense, the presented material is not an example of typical advanced oxidation. However, considering the purpose of advanced oxidation and advanced technology, we can state the similarity of this process with the presented advanced biological oxidation. The main purpose of advanced oxidation in wastewater treatment is the removal of persistent organic compounds, traceable emerging contaminants and certain inorganic contaminants [34]. Typical domestic wastewater consists not only of easily degradable organic compounds such as fibers, proteins and sugars, which account for approximately 45% of the total chemical oxygen demand (COD) in raw wastewater [35], but also many other contaminants derived from soaps, shampoos, conditioners, personal care products, detergents and laundry products [36]. These substances, which may be more difficult to degrade, together with inorganic materials, are responsible for the remaining oxygen demand in raw wastewater. Treatment processes employed in conventional wastewater treatment plants are not efficient in removing all impurities from the wastewater [37].

To reuse treated wastewater, advanced technologies such as advanced oxidation must be used. This allows us to obtain safer water than from a typical wastewater treatment plant. In order to verify the compliance of the presented results with the requirements of advanced oxidation and advanced technology, it is necessary to consider the final effect of these processes. Table 4 shows the concentration of BOD and ammonium nitrogen (NH$_4$-N) in the effluent from compost beds A, B and C. To evaluate this oxidation effect presented in this table, it is necessary to compare the results with the requirements regarding the concentration of BOD in the effluent of conventional treatment plants (Table 5). This comparison allows us to state that the effluent of each compost bed significantly exceeds the requirements regarding BOD by all sizes of conventional secondary treatment. As shown in the previous study [13], the elimination of COD in this plant was equally effective and exceeded all requirements for wastewater treatment plants of all sizes. The average COD values in the effluent from compost beds A, B and C were 61.1 mg/L, 72.0 mg/L and 85.9 mg/L, respectively (see Table 2 in [13]). Furthermore, these values were characterized by a slight downward trend throughout the study period (see Figure 3 in [13]).

Furthermore, comparing the obtained results with the effects of wastewater treatment with advanced technology (Table 6), it can be seen that BOD and NH$_4$-N concentrations in both approaches are very similar. These results confirm that such a high degree of BOD treatment is possible in conventional municipal wastewater treatment plants only with the use of additional polishing processes, such as flotation and membrane filtration [38]. The high CV values in the effluent are caused by low absolute concentration values. These can vary by as much as 100% but are still low and do not, for example, exceed the limits for safe agricultural use of the treated sewage.

**Table 4.** Average concentration of biological oxygen demand and ammonium nitrogen in compost beds.

| | | A—0.07 m$^3$/m$^2$ | | B—0.1 m$^3$/m$^2$ | | C—0.13 m$^3$/m$^2$ | |
| | | BOD | NH$_4$-N | BOD | NH$_4$-N | BOD | NH$_4$-N |
|---|---|---|---|---|---|---|---|
| Influent | (g O$_2$/m$^3$) | 232.1 | 135.9 | 232.1 | 135.9 | 232.1 | 135.9 |
| | SD | 91.9 | 52.9 | 91.9 | 52.9 | 91.9 | 52.9 |
| | CV | 40.0 | 38.9 | 40.0 | 38.9 | 40.0 | 38.9 |
| Effluent | (g O$_2$/m$^3$) | 4.0 | 2.0 | 5.4 | 4.3 | 7.2 | 4.2 |
| | SD | 1.8 | 1.4 | 2.9 | 4.9 | 4.9 | 2.7 |
| | CV | 44.2 | 66.6 | 53.7 | 113.2 | 71.3 | 65.1 |
| Reduction | (%) | 98.3 | 98.5 | 97.7 | 96.8 | 96.9 | 96.8 |

**Table 5.** The highest permissible values of pollution indicators or the minimum percentages of pollution reduction for domestic and municipal wastewater treatment plants discharged into waters and soils in Poland [39]. N/A—not available.

| Parameter | Unit | Treatment Plant Size (Number of Inhabitants) | | | |
|---|---|---|---|---|---|
| | | <2000 | 2000–9999 | 10,000–14,999 | 15,000–99,999 |
| BOD | g $O_2$/m$^3$ | 40 | 25 | 25 | 15 |
| | (% reduction) | (N/A) | (75–90) | (70–90) | (90) |
| COD | g $O_2$/m$^3$ | 150 | 125 | 125 | 125 |
| Total | g $O_2$/m$^3$ | 30 | 15 | 15 | 15 |
| Nitrogen | (% reduction) | (N/A) | (N/A) | (70–80) | (70–80) |

**Table 6.** Effluent quality of selected advanced wastewater treatment plants in California [40].

| Quality Parameter (g $O_2$/m$^3$) | Long Beach | Los Coyotes | Pomona | Plant Location Dublin San Ramon | Livermore | Simi Valley CSD | This Study |
|---|---|---|---|---|---|---|---|
| BOD | 5.0 | 9.0 | 4.0 | 2.0 | 3.0 | 4.0 | 5.5 |
| $NH_4$-N | 3.3 | 13.6 | 11.4 | 0.1 | 1.0 | 16.6 | 3.5 |

The compost beds not only provide ammonium oxidation at levels similar to conventional advanced technologies but also provide very effective removal of total nitrogen, averaging 68% for all beds [14]. This removal efficiency is similar to the requirements for wastewater treatment plants (Table 5). Another way to evaluate the BOD concentration in the compost bed effluent is to compare it with water reuse requirements. In the European Union (EU), the highest requirements for water quality class A (all food crops consumed raw where the edible part is in direct contact with reclaimed water, as well as root crops consumed raw) indicate a BOD concentration of less than 10 g $O_2$/m$^3$ [41]. The same requirements for BOD (<10 g $O_2$/m$^3$) in reclaimed water apply in the USA and Canada [42,43]. The safety of reusing the treated wastewater after compost beds is also confirmed by an average pH of 6.17 and an average electrical conductivity of 1535 µS/cm from the compost bed effluent. Considering that the effluent from three compost beds (1) meets the highest requirements for water reuse, (2) is similar to the concentration of BOD and $NH_4$-N in the effluent from advanced technology wastewater treatment and (3) significantly exceeds the requirements for BOD of conventional secondary treatment, it can be concluded that the results of the present study are a very good example of advanced biological oxidation and advanced technology.

Another important factor confirming the successes of full biological oxidation is the content of oxygen in the effluent of each compost bed (Figure 5). Oxygen was available during the study period for all processes in compost beds (see Results). Moreover, it is not a limiting factor for the purification processes, which very often happens with conventional NTSWs. In addition, it is worth paying attention to the fact that oxygen supply in compost beds ensured available oxygen not only for heterotrophic and autotrophic bacteria but also for other living organisms in compost beds. On the other hand, the very high oxygen concentration in the effluents (average value of 7 g $O_2$/m$^3$) should be emphasized. First of all, the dissolved oxygen content in the effluents of wastewater treatment plant processes reflects the efficiency of the biological treatment phase [44]. The higher the oxygen concentration in the effluent, the better the operation of the wastewater treatment plant. Secondly, when discharging effluent to surface water, it is essential to ensure that the dissolved oxygen concentration in the effluent is at least 4 g $O_2$/m$^3$. Below this level, aquatic organisms are adversely affected [44]. Moreover, the concentration of dissolved oxygen in the effluent from each compost bed fluctuated within the range corresponding to the first and second class (cleanest water) of surface water quality assessment in Poland and the EU [45], which indicates a very good oxygen condition in the reclaimed water.

The low hydraulic retention time also confirms that the technology presented could be classified as advanced biological oxidation and advanced technology. In conventional NTSWs such as wetland systems, the hydraulic retention time ranges from about 1 to 15 days [23]. In conventional treatment systems, such as activated sludge, which is the most popular biological treatment process, the HRT typically varies from 1.5 to 24 h [46]. Using activated sludge with membrane filtration (advanced technology), the HRT can vary from 13 to 19 h [47]. On the other hand, the HRT in the compost beds of the presented pilot plant is only about 1 h. This means that during 1 h, all organic substances in wastewater are decomposed, and almost all ammonium nitrogen is oxidized simultaneously. These facts further enhance the statement that from a technological point of view, presented solutions based on compost beds can be treated as advanced biological oxidation and advanced technology. However, considering that (1) the whole advanced biological process takes place without additional energy supply, (2) the process is running without special technical equipment, and (3) no control process is required, from a technical point of view, the presented technology can also be considered an example of low technology.

The NTSW plant presented in this study also provides effective removal of microbiological contaminants. Conventional wastewater treatment plants are designed to remove suspended solids, organic matter and nutrients but do not emphasize the removal of microbiological contaminants [48]. Reuse of water in agriculture requires disinfection of the effluent from conventional wastewater treatment plants. Each of the disinfection processes used is considered an advanced process [49]. According to the UE requirements for wastewater reuse in agriculture, the effluent from the conventional treatment system must be disinfected by additional processes to achieve the following reuse water classes: A (*E.coli* $\leq$ 10 cfu/100 mL), B (*E.coli* $\leq$ 100 cfu/100 mL), C (*E.coli* $\leq$ 1000 cfu/100 mL) and D (*E.coli* $\leq$ 10,000 cfu/100 mL) [41].

The concentration of *E.coli* in the effluent of all the samples taken from compost bed A is within the range for classes B and C. This fact is an additional confirmation that the purification process of wastewater treatment in compost beds is an example of advanced biological oxidation and advanced technology because this process includes not only the removal of suspended solids, organic matter and nutrients but also disinfection. The effectiveness of disinfection in the compost beds could be concluded by comparing the *E. coli* and enterococci bacteria in the effluent of compost bed A with the EU requirements for bathing water. According to these requirements, the residual bacteriological contamination in the effluent meets the strictest requirements for excellent bathing water quality in inland and transitional waters [32]. This also ensures that water reuse after composting is safe and does not pose a risk to humans, animals or the environment.

A final aspect confirming the high efficiency of advanced biological oxidation in compost beds is to compare the efficiency of the beds themselves to that of the overall NTSW studied. As mentioned in Section 2.1 (Description of the NTSW Pilot Plant), the aim of the project was to develop an NTSW, consisting of compost filters and renewal beds, which would provide treatment of domestic wastewater to drinking water quality. The NTSW pilot plant provided full removal of both readily degradable organic matter (99.1%) and ammoniacal nitrogen (99.5%). If we compare the efficiency of pollutant elimination in the compost beds with that in the renewal beds (Table 7), it is clear that in the entire process of treating domestic wastewater to drinking water quality, as much as 97.5% of the BOD reduction, 82% of the COD reduction and 97.5% of the ammoniacal nitrogen reduction occurs in the compost beds. In addition, the compost beds provided a total reduction of up to 68% in total nitrogen. The presented results confirm the high efficiency of the biological oxidation process in compost beds. Without their operation, it would be impossible to treat domestic wastewater to a quality close to drinking water.

The presented technology of using composting beds as a purification bioreactor allows the simultaneous production of good-quality compost. Since all these processes take place in a bioreactor (compost bed) with sufficient oxygen supply not only for bacteria but also for other different organisms responsible for wood chip mineralization and compost

production, this technology could be considered advanced biological oxidation and advanced technology, the final effect of which is much more effective and diverse than that of conventional treatment systems. The compost produced by this advanced technology has an additional purification effect. The nutrient composition of this compost is similar to traditional compost and manure from different types of livestock [32,50,51].

**Table 7.** Quality parameters of the raw sewage, effluent of the compost beds and water renewal beds, achieved in the studied NTSW pilot plant (based on Tables 1 and 2 from [13] and Tables 2 and 3 from [14]).

| Quality Parameter (mg/dm$^3$) | BOD | COD | NH$_4$-N | NO$_3$ | TN |
|---|---|---|---|---|---|
| Raw sewage | 232.1 | 407.3 | 135.9 | 0.1 | 130.7 |
| Effluent of the compost beds | 5.6 | 73.0 | 3.5 | 132.1 | 42.2 |
| Effluent of the water renewal beds | 2.2 | 17.8 | 0.6 | 49.4 | 12.0 |

This additional compost production has many environmental benefits during its use. According to various studies, the use of compost as a soil amendment improves soil quality, including (1) incorporating organic matter, nutrients and electrolytes into the soil, (2) reducing the need for fertilizer, pesticides and peat use, (3) improving soil structure, density and porosity, which increases water-holding capacity and reduces erosion and nutrient leaching, and (4) increasing the carbon storage capacity of the soil, thereby reducing global warming [52,53].

## 5. Conclusions

Based on the results of this study, it could be stated that not only technical methods but also some types of natural treatment systems are capable of advanced wastewater treatment. The presented method using compost beds as a bioreactor for the wastewater treatment process enables full biological oxidation for almost all organic substances as well as for ammonium nitrogen in the range of 96.8 to 98.5%. Due to the lower daily hydraulic load of compost bed A, which is 0.07 m$^3$/m$^2$, the average concentration of BOD in the effluent of this bed is only 4 g O$_2$/m$^3$ for the whole sampling period. This result is similar to the BOD concentration in clean surface water and the effluent from wastewater treatment plants with additional advanced technology. Both parameters responsible for oxygen demand in wastewater (BOD and NH$_4$-N) in the effluent from all beds meet the highest requirements for recycled water used for irrigation purposes in the EU, USA and Canada. The safety of water reuse is also confirmed by the high removal efficiency in terms of bacteriological contamination (256 cfu/100 mL and 61 cfu/100 mL for the *E. coli* and enterococci in the effluent, respectively). The structure of the compost beds provides very good conditions for oxygen supply to heterotrophic and autotrophic bacteria and higher organisms living in them. Moreover, such oxygen condition allows oxygen saturation in the recycled water to the level corresponding to the cleanest surface water. This fact confirms that during the whole year, the oxygen demand in the wastewater was covered to 100% in the compost beds. The oxidation of organic matter and ammonium nitrogen, as well as the saturation of the treated water in the compost beds, took place in a very short hydraulic residence time, which was only about 1 h. All the above facts confirm that the presented method of natural treatment systems using compost as a biological reactor can be considered advanced biological oxidation and advanced technology. Taking into account all these advantages, as well as the high removal efficiency of COD and TN (82% and 68%, respectively) in these compost beds [13,14], it can be concluded that the presented method is a good example of full local wastewater treatment and reuse of reclaimed water. This method is characterized by low construction and operating costs and additionally enables the simultaneous production of good-quality compost. In Poland, the investment and operating costs of such an advanced treatment plant are similar to those of conventional treatment plants of comparable size. Such a method could be a good alternative to conventional advanced methods for households and small economies.

Currently, examples of this system have been used in Poland as an advanced technology for wastewater treatment and water reuse for two schools in western Poland: a public school in Wężyska, Krosno Odrzańskie community (250 pupils), and a public school in Szczawno, Dąbie community (100 pupils). Further, such NTSW plants have been installed at 200 households in the Korytnica community (eastern Poland).

**Funding:** This research was funded by the National Center for Research and Development, grant number POIR.01.01.01-00-0805/18.

**Data Availability Statement:** Not applicable.

**Acknowledgments:** The author acknowledges the support of Michał Halicki and Nerie Taberna in the article production process.

**Conflicts of Interest:** The author declares no conflict of interest. The funders had no role in the design of the study; in the collection, analyses or interpretation of data; in the writing of the manuscript or in the decision to publish the results.

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
