# Peer review of "Advanced Biological Oxidation of Domestic Sewage with the Use of Compost Beds in a Natural Treatment System for Wastewater"

_sustainability, doi:10.3390/su151813555_

Round 1

Reviewer 1 Report

The author has improved the paper according to the previous comments. In order to improve the discussion of the results, the author could add a comparative table of the treatment performances of this composting filter and other conventionnel biofilters to demonstrate the effectiveness of the proposed treatment system.

Minor editing of English language required

Author Response

The author has improved the paper according to the previous comments. In order to improve the discussion of the results, the author could add a comparative table of the treatment performances of this composting filter and other conventionnel biofilters to demonstrate the effectiveness of the proposed treatment system.

 I thank the Reviewer for this suggestion – I also think that such a table is of great value for this study. I have already presented such a table in the Discussion (Table 6), where I present the effluent quality of selected advanced wastewater treatment plants in California. However, this table may have lacked a direct comparison with the results of this study, so I have added a column “This study” with the average BOD and NH4-N concentrations. This makes it easier to compare the performance of the treatment plants. It clearly shows that my results are comparable with other conventional advanced treatment plants. In addition, I have enriched the discussion with another citation which claims that such high removal efficiencies are only possible in conventional municipal wastewater treatment plants only with the use of additional polishing processes, such as flotation and membrane filtration (Discussion, lines 375-378).

Reviewer 2 Report

The author states that during this 2.5-year research study, wastewater and reclaimed water were analyzed for the following parameters: Biological Oxygen Demand (BOD), Chemical Oxygen Demand (COD), oxidation, Total Sus-pended Solids (TSS), Total Nitrogen (TN), NH4, NO3, NO2, Total Phosphorus (TP), PO4, pH, conductivity, oxygen, K, Cl, Mn, Ca, Fe, coli bacteria, enterococci, and colony count 22°C. Unfortunately, there are no test results data in the work. Based on the presented results, it cannot be concluded that the purification is taking place correctly. In the system under assessment, NH4-N oxidation occurs, but TN is at a constant level. Recirculation of such wastewater will increase the concentration of TN. The author does not provide a solution to this problem. Microbiological analysis is not transparent, methodology and test results are missing. The text correlates poorly with figure 6.

In table 1 The author gives oxygen demand in raw wastewater flowing from the septic tank to compost beds for NH4-N. How did he calculate the given values? Please explain.

The work still requires additional research, presentation of the results of the analyzes carried out. Currently, it is not convincing material.

Author Response

The author states that during this 2.5-year research study, wastewater and reclaimed water were analyzed for the following parameters: Biological Oxygen Demand (BOD), Chemical Oxygen Demand (COD), oxidation, Total Sus-pended Solids (TSS), Total Nitrogen (TN), NH4, NO3, NO2, Total Phosphorus (TP), PO4, pH, conductivity, oxygen, K, Cl, Mn, Ca, Fe, coli bacteria, enterococci, and colony count 22°C. Unfortunately, there are no test results data in the work. Based on the presented results, it cannot be concluded that the purification is taking place correctly.

Following the suggestion of an earlier reviewer, I have included in this article a list of all the analyses that were performed during the research project, within which this study has been conducted. However, due to the large scope of the research material and the multi-faceted nature of the project, I have decided to spread the publication of the results over several articles. In the Description of the NTSW Pilot Plant section, I have clearly stated that I have already published the first part of the results dealing with the removal of organic matter. I have also cited this article. The second part of the results concerning the nitrogen and phosphorus removal aspect has also been published and I have given the literature data of this publication in the description. So far, no one has questioned that I only use part of the results in the individual articles to discuss a selected aspect of the research. This article only focuses on the biological advanced oxidation process. If the reader would like to know the rest of the results, he can look to the previous two articles, which are available in open access. These articles, together with the published results in this article, represent the entire project.

In the system under assessment, NH4-N oxidation occurs, but TN is at a constant level. Recirculation of such wastewater will increase the concentration of TN. The author does not provide a solution to this problem.

I already stated in the discussion (lines 391-394) that compost filters provide a TN reduction of 68%, therefore I cannot agree with the Reviewer’s statement, that TN is at a constant level. Here I also cited the literature, i.e. the other article, which was precisely on the issue of removal of nitrogen compounds, including TN, from wastewater. I do not quite understand the remark about recirculation, as recirculation did not take place in the system under study.

Microbiological analysis is not transparent, methodology and test results are missing. The text correlates poorly with figure 6.

I have followed the Reviewer’s suggestion and made the following corrections:

  • In section 3.2. (lines 307-309) I have added the mean, minimum and maximum values for E.coli and enterococci. I believe that these values, together with Figure 6 showing the distribution of these results in relation to the European requirements, are sufficient.
  • I have made corrections to the text in section 3.2. (lines 309-321), which I believe improve the readability of Figure 6 and its correlation with the text.

Also, in the section 2.2. I have already presented the methods (ISO standards), the sampling location (effluent of the compost filter A), the number of samples taken, and the time period within which the sampling took place. I hope that the improved results, the corrected description of the Figure 6, and the description of sampling presented above are sufficient and will satisfy the Reviewer.

In table 1 The author gives oxygen demand in raw wastewater flowing from the septic tank to compost beds for NH4-N.
How did he calculate the given values? Please explain.

These values were calculated from the stoichiometric equations presented by Rheinheimer et al (1988). These equations have been added to the description of the test results (Results, lines 239-242), I also present them below:

NH4+ + 1.5O2  2H+ + H2O + NO2
3.21 mg O2 / mg NH4+-N,

(1)

NO2 + 0.5O2  NO3,
1.11 mgO2 / mg NO2-N,

(2)

Rheinheimer, G. Stickstoffkreislauf im Wasser: Stickstoffumsetzungen in Natürlichen Gewässern, in der Abwasserreinigung und Wasserversorgung (Nitrogen cycle in water: nitrogen transformations in natural waters, in wastewater treatment and water supply); R. Oldenbourg Verlag: Munich, Germany, 1988; ISBN 978-3-486-26296-4.

The work still requires additional research, presentation of the results of the analyzes carried out. Currently, it is not convincing material.

The article was further enhanced by comments of other Reviewers, for example by (1) providing standard deviations and coeficients of variation of the BOD and NH4-N concentrations, (2) by providing the total number of samples taken from the treated and raw wastewater and (3) by refining Table 6 and adding a new reference which should improve the discussion and comparison with other treatment plants. I hope that the changes made at the request of the Reviewer (as well as the changes made at the request of the other reviewers) and clarifications have improved the quality of this article and will satisfy the Reviewer.

Reviewer 3 Report

Dear Author,

the manuscript addresses the important topic of Advanced Biological Oxidation of Domestic Sewage. However, due to errors in the description of the methodology and lack of statistics, it must be sent for minor revision.

Please see detailed notes below.

Methods - Sampling and Quality Assessment

How many total samples of raw and treated sewage were there in terms of physicochemical and bacteriological analyses?

Results

Did the treated wastewater meet the quality criteria for treated wastewater used for irrigation in terms of bacteriology?

The manuscript lacks a table similar to Table 4 with E. coli and enterococci bacteria reduction rates and basic descriptive statistics (as above).

Notes on statistics

3.1. Oxygen Demand

Please add the standard deviation and coefficient of variation for the measured oxygen values and its reductions (Table 1 and Table 2)

Please provide the standard deviation and the coefficient of variation for the measured BOD and ammonium nitrogen concentrations and the degree of their reduction (Table 4).

Author Response

Dear Author,

the manuscript addresses the important topic of Advanced Biological Oxidation of Domestic Sewage. However, due to errors in the description of the methodology and lack of statistics, it must be sent for minor revision. Please see detailed notes below.

Methods - Sampling and Quality Assessment

How many total samples of raw and treated sewage were there in terms of physicochemical and bacteriological analyses?

Sampling is described in section 2.2. I have given the time range of measurements and the frequency (approximately every 2 weeks for treated wastewater and approximately every month for raw sewage). I have also given the time range and number of bacteriological measurements (16 and 15). I have also added a sentence with the number of raw and treated wastewater samples, which is 48 and 92 for the raw and treated wastewater, respectively (section 2.2., lines 163-166).

Results

Did the treated wastewater meet the quality criteria for treated wastewater used for irrigation in terms of bacteriology?

The answer to this question is already presented in the discussion, lines 446-452. In this case, the outflow from the compost filters meets the criteria for treated effluent used for irrigation in the range of Class B and C according to EU standards.

The manuscript lacks a table similar to Table 4 with E. coli and enterococci bacteria reduction rates and basic descriptive statistics (as above).

I agree that such a table would have been an added value to this article, but unfortunately the bacteriological studies were not carried out on raw wastewater. Therefore, it is not possible to give the percentage of reduction, but only the number of bacteria in the treated wastewater. This number meets the requirements of good quality bathing water and agricultural irrigation water (see Discussion). It also meets the requirements for the B and C class for treated effluent used for irrigation in EU, as mentioned in the previous answer.

Notes on statistics

3.1. Oxygen Demand

Please add the standard deviation and coefficient of variation for the measured oxygen values and its reductions (Table 1 and Table 2)

I have supplemented the tables with the indicated statistics. I have also added a short commentary to the results next to each table (lines 246-248 for Table 1 and lines 267-270 for Table 2).

Please provide the standard deviation and the coefficient of variation for the measured BOD and ammonium nitrogen concentrations and the degree of their reduction (Table 4).

I have completed the tables with the indicated statistics. I have also added a short commentary on the results next to this table (lines 378-380). Nevertheless, I have not given the SD and CV for the reduction because the raw and treated effluent measurements were carried out at different time intervals, so only the average reduction over the entire study period is presented. Nevertheless, providing the SD and CV for influent and effluent helps to illustrate the variability of the parameters and gives an overall view of the possible variations in reduction. Regardless of the SD and CV values for reduction, in this article I have shown that the described plant provides low concentrations of BOD and NH4-N in the effluent and high reduction efficiency. It can therefore be concluded that the presented method is a good example of full local wastewater treatment and reuse of reclaimed water.

Round 2

Reviewer 2 Report

The author states th during this 2.5-year research study, wastewater and reclaimed water were analyzed for the following parameters: Biological Oxygen Demand (BOD), Chemical Oxygen Demand (COD), oxidation, Total Sus-pended Solids (TSS), Total Nitrogen (TN), NH4, NO3, NO2, Total Phosphorus (TP), PO4, pH, conductivity, oxygen, K, Cl, Mn, Ca, Fe, coli bacteria, enterococci, and colony count 22°C. Unfortunately, there are no test results data in the work. Based on the presented results, it cannot be concluded that the purification is taking place correctly. In the system under assessment, NH4-N oxidation occurs, but TN is at a constant level. Recirculation of  wastewater will increase the concentration of TN. The author does not provide a solution to this problem. Microbiological analysis is not transparent, methodology and test results are missing. The text correlates poorly with figure 6.
In table 1 The author gives oxygen demand in raw wastewater flowing from the septic tank to compost beds for NH4-N. How did he calculate the given values? Please explain.

The work still requires additional research, presentation of the results of the analyzes carried out. Currently, it is not convincing material.
If, in accordance with the title of the work, the author dealt only with biological oxidation. There is too little interpretation of the process, there is a lack of microbiological analysis and theoretical description. The effects of purification cannot be assessed based on the result data. Existing research should be justified by a full analysis. These results are good but have no practical justification.

The method of wastewater treatment used may be good, but the justification is poor.

The author should disclose more research results, which he informs about in the abstract

Round 3

Reviewer 2 Report

* Please confirm that you appreciate to be notified with the final status of the paper